# Genetic Diversity and Phylodynamics of Avian Coronaviruses in Egyptian Wild Birds

**DOI:** 10.3390/v11010057

**Published:** 2019-01-14

**Authors:** Mohammed A. Rohaim, Rania F. El Naggar, Ahmed M. Helal, Mahmoud M. Bayoumi, Mohamed A. El-Saied, Kawkab A. Ahmed, Muhammad Z. Shabbir, Muhammad Munir

**Affiliations:** 1Virology Department, Faculty of Veterinary Medicine, Cairo University, Giza 12211, Egypt; mohammed_abdelmohsen@cu.edu.eg (M.A.R.); mahmoud.bayoumi@cu.edu.eg (M.M.B.); 2Division of Biomedical and Life Sciences, Faculty of Health and Medicine, Lancaster University, Lancaster LA1 4YG, UK; 3Virology Department, Faculty of Veterinary Medicine, Sadat University, Sadat 32897, Egypt; raniamohammedvet@gmail.com; 4Central Lab for Evaluation of Veterinary Biologics, Abbasia 11381, Cairo, Egypt; ahmedmahervet2020@gmail.com; 5Pathology Department, Faculty of Veterinary Medicine, Cairo University, Giza 12211, Egypt; vet.mohamedsaied@gmail.com (M.A.E.-S.); kawkababdelaziz@cu.edu.eg (K.A.A.); 6Quality Operations Laboratory, University of Veterinary and Animal Sciences, Lahore 54600, Pakistan; shabbirmz@uvas.edu.pk

**Keywords:** avian coronavirus, Egypt, wild bird, phylodynamics, monitoring

## Abstract

Avian coronaviruses (ACoVs) are continuously evolving and causing serious economic consequences in the poultry industry and around the globe. Owing to their extensive genetic diversity and high mutation rates, controlling ACoVs has become a challenge. In this context, the potential contribution of wild birds in the disease dynamics, especially in domesticated birds, remains largely unknown. In the present study, five hundred fifty-seven (*n* = 557) cloacal/fecal swabs were collected from four different wild bird species from eight Egyptian governorates during 2016 and a total of fourteen positive isolates were used for phylodynamics and evolutionary analysis. Genetic relatedness based on spike (S1) gene demonstrated the clustering of majority of these isolates where nine isolates grouped within Egy/variant 2 (IS/885 genotype) and five isolates clustered within Egy/variant 1 (IS/1494/06 genotype). Interestingly, these isolates showed noticeable genetic diversity and were clustered distal to the previously characterized Egy/variant 1 and Egy/variant 2 in Egyptian commercial poultry. The S1 gene based comparison of nucleotide identity percentages revealed that all fourteen isolates reported in this study were genetically related to the variant GI-23 lineage with 92–100% identity. Taken together, our results demonstrate that ACoVs are circulating in Egyptian wild birds and highlight their possible contributions in the disease dynamics. The study also proposes that regular monitoring of the ACoVs in wild birds is required to effectively assess the role of wild birds in disease spread, and the emergence of ACoVs strains in the country.

## 1. Introduction

Coronaviruses are classified into three well-accepted genera; alpha-, beta-, and gamma-coronavirus whereas deltacoronavirus [1,2] is proposed as an additional group in the family. Alphacoronaviruses and betacoronaviruses have been isolated from mammals, whereas gammacoronaviruses, represented by the infectious bronchitis virus (IBV), are detected primarily in birds including poultry [2].

Avian coronvaviruses (ACoVs) are enveloped viruses with a positive-sense and single-stranded RNA genome of approximately 27.6 kb in length that encodes four major structural proteins; the nucleocapsid (N) protein, the membrane (M) protein, the envelope (E) protein and the spike (S) glycoprotein. The S glycoprotein, an integral membrane protein, is post-translationally cleaved into the S1 (N-terminus) and S2 subunits (C-terminus) [3]. The S1 protein is responsible for host selection, induction of protective immunity and most of the neutralizing serotype-specific antibodies are directed against S1 [4,5]. Most of these neutralizing epitopes are mapped in the hypervariable region (HVR) in the S1 subunit (amino acid positions from 38–67, 91–141 and 274–387) and mutation in these regions may contribute to the generation of escape mutants [5]. Since the S1 subunit shows higher sequence diversity (up to 50%) than S2 [6], molecular characterization and serotyping of coronaviruses are based mainly on the S1 gene sequences [7]. The S1 gene sequence can be used to differentiate all six IBV genotypes, and 32 different lineages that have been identified worldwide [8]. New genotypes of IBV are emerging frequently in different parts of the world and many factors are accounting for this emergence and evolution [9].

ACoV infections are characterized by an acute, highly contagious, and economically important disease in domesticated poultry. After its first identification in North Dakota, USA [10], ACoV spread swiftly and many ACoV variant strains have been isolated causing major problems in the poultry industry, around the globe [3,7,8,9,11,12]. Contaminated litter, footwear, clothing, utensils, equipment, and personnel are all potential sources of virus for indirect transmission and have been implicated in ACoV spread over large distances [12]. However, the role of wild birds in disease transmission, pathobiology, and evolution of the virus is not well understood.

The wildlife has been under epidemiological surveillance to identify its possible roles as a reservoir for emerging viruses that may pose a risk to mankind and threaten wildlife. It has now been well established that the wild birds are important reservoirs for avian influenza A virus [13], and may likely be for other respiratory and enteric viruses. In recent years, ACoVs have been isolated from various wild bird species that create a suspicion that wild birds play a role as CoV reservoirs and as ACoV carrier/transmitter influencing its epidemiology [14,15]. In Egypt, ACoV was first described at 1954 by Ahmed, [16], and subsequently several reports emphasized the prevalence of the disease in the country that results in great economic losses to the Egyptian commercial poultry industry [17,18,19]. Currently, the variant IBV are the most prevalent strains in commercial poultry sectors in Egypt, especially from the early months of 2011. Therefore, genotyping of ACoV field strains is crucial to detect the emergence of new variant strains as well as evaluating the existing vaccination programs [19]. To date, only limited studies have been performed to systematically monitor the prevalence of ACoVs in wild bird populations.

The main objective of this study was to investigate the prevalence of ACoVs in the Egyptian wild birds and to highlight their possible roles in transmission of ACoVs. Screening and downstream characterization and evolutionary analysis confirm the circulation of ACoVs in wild birds and warrant the future surveillance investigations at the national levels. Understanding dynamics and richness of Egyptian wilds birds in carrying avian pathogens will be steppingstone in devising any disease control and eradication campaign in the country.

## 2. Materials and Methods

### 2.1. Sampling History, Collection, and Virus Isolation

A total of five hundred- fifty-seven (*n* = 557) cloacal/fecal swabs were collected from randomly-selected wild birds-dense and ACoV-endemic areas in eight Egyptian provinces during the year of 2016 (Figure 1, Table 1, Table 2, Table 3). Swab samples from live birds were obtained according to approved procedures by authorized veterinarians of the Central Laboratory for Evaluation of Veterinary Biologics, Egypt. Samples were blindly propagated for three passages in the allantoic sac of 9 day-old specific pathogen free (SPF) embryonated chicken eggs by standard procedures [20].

### 2.2. Polymerase (RdRp) and Spike (S1) Genes Amplification and Sequencing

Viral RNA was extracted from the allantoic fluids using Trizol LS® reagent kit (ThermoFisher, Life Technologies Corporation|Carlsbad, CA 92008, USA) according to the manufacturer’s instructions. The eluted RNA in 30 µL of nuclease free water was stored at (−70 °C) until processing for the amplification. A single step RT-PCR was performed using the Verso One Step RT-PCR kit (ThermoFisher, Life Technologies Corporation|Carlsbad, California, USA) using previously reported primers for amplification of partial polymerase (RdRp) and full length S1 gene [6,21]. PCR products were gel excised and purified individually using QIAEX^®^ Gel Extraction Kit (Qiagen, Hilden, Germany). Gene sequencing was carried out using a BigDye Terminator v3.1 cycle sequencing kit (Qiagen, Hilden, Germany) in an ABI PRISM® 300 using sequencing primers as described previously [22].

### 2.3. Sequence and Phylogenetic Analysis

The obtained sequences for RdRp (MF034358.1- MF034371.1) and S1 genes (MF034369.1- MF034385.1) were quality checked, assembled, edited using BioEdit software version 7.0.4.1 [23] and submitted to GenBank using BankIt tool of the GenBank (http://www.ncbi.nlm.nih.gov/WebSub/?tool=genbank). Analysis of full length S1 protein was conducted to determine the presence of potential N-linked glycosylation sites among the isolated Egyptian ACoV strains using BioEdit software version 7.0.4.1 [23]. To elucidate the phylogenetic relationships and high level clustering pattern based on full length S1 gene between ACoVs reported here and previously characterized from Middle East including Egypt and other parts of the world; we first compiled a dataset of all available S1 gene sequences. Then, this dataset was aligned in BioEdit version 7.0.1 [23] using Clustal W algorithm and was trimmed to equal lengths. To avoid sequence length biasness in evolutionary incursion, all sequences that aligned poorly or with incomplete information were excluded from the analysis. A phylogenetic tree was constructed using the nucleotide sequences of the S1 genes. For maximum-likelihood analysis of the phylogenetic relationship, a best-fit model was chosen on which further calculations and an ultrafast bootstrap equivalent analysis were based. IQ-tree software version 1.1.3 was used for these operations [24].

### 2.4. Selection Pressure Analysis 

To explore the overall differences in selection pressure on the S1 gene, especially on the hypervariable regions (HVRs) sequences that define the cross-neutralization and escape mutant ACoV strains, we analyzed the occurrences of synonymous (dS) and non-synonymous (dN) substitutions using SNAP web tool (https://www.hiv.lanl.gov/content/sequence/SNAP/SNAP.html) [25], which plots the cumulative and per-codon occurrence of each substitutions. Six methods (RDP, GeneConv, BootScan, MaxChi, Chimaera and SiScan) integrated in the RDP v3 program [26] were applied on the avian CoVs sequences reported here to estimate any recombination event and to detect any putative recombination breakpoint using the nucleotide alignment of the S1 gene sequences. These methods were applied using the following parameters: window size = 20, highest acceptable *p*-value = 0.001 and Bonferroni correction. For reliable results, any putative recombination events detected by more than one method were considered.

## 3. Results

### 3.1. Occurrence and Distribution of Avian Coronavirus in Wild Birds

A total of 557 samples were collected from randomly selected wild birds dense and ACoV endemic areas from eight provinces. These samples were individually screened by the RT-PCR targeting the RdRp gene followed by the full length S1 gene of the ACoVs. From this screening, 2.5% positive rate (14 out of 557) was detected among all tested fecal/cloacal samples. We classified all species that were included in the analysis into four different families that reflected both their taxonomy and their ecology (Table 1). These families were Passeridae (1 species, *n* = 140), Phasianidae (1 species, *n* = 136), Anatidae (1 species, *n* = 150) and Ardeidae (1 species, *n* = 131). Anatidaeily was most frequently noted as ACoV-positive (4.7%), followed by Ardeidae (3.1%) and Phasianidae (1.5%), while Passeridae had the lowest prevalence (0.7%). Furthermore, there were notable variations in the proportion of positive samples between provinces: Sharqia, Dakahlia, Kafr El Sheikh, Gharbia, Qalubia, Menofia and Benisuef (Table 2). The lowest prevalence rates were observed on Dakahlia (1.2%) and Menofia (1.7%), whereas Qalubia (2.5%), Gharbia (2.4%), Benisuef (2.3%) were at the mid-point, and the highest rates were detected at Kafr El Sheikh (5%) and Sharqia (3.3%) (Table 2). However, Giza province did not show any positive ACoV (Table 2).

### 3.2. Viral Sequences and Genomics Analysis

Comparison of nucleotide identity percentages, based up on S1 gene revealed that the fourteen isolates were 92–100% identity percentage (Table 3). In 2001, an Egyptian variant I strain (Egypt/Beni-Suef/01), closely related to the Israeli variant strain, was reported in different poultry farms [18]. Another variant was also identified in 2011 [27] and designated as Egyptian variant II. Deduced amino acids analysis was conducted to establish the genetic spectrum, origin, evolution, and the mutation trend analysis for Egyptian ACoVs. Hypervariable regions (HVRs) in the S1 gene showed distinct patterns among different viruses compared to the H120 and Ma5 strains (Table 4). In the analyzed sequence of the fourteen viruses in this study, amino acid mutations associated with virus tropism were identified at positions 38 (38N in only one virus while 38T in thirteen viruses) and 69 (69S in seven viruses and 69A in seven viruses) in the S1 protein of the variant strains. In addition, seventeen potential N-linked glycosylation sites in most of the Egyptian ACoV strains were found, with nine NXT and eight NXS sites in all the fourteen isolates reported in this study.

### 3.3. Phylogenetic Analysis

Based on the amplified CoV polymerase (RdRp) gene fragments, the phylogenetic analyses confirmed that the all detected viruses in this study belonged to gamma-coronaviruses (Figure 2) along with other ACoVs. Phylogenomic analysis based on S1 gene of the ACoVs collected from Egypt and rest of the world, indicated that Egyptian viruses reported here were clustered with isolates from Libya, Oman and Kurdistan and shared ancestors (Figure 3, radiation tree, left panel). A higher resolution analysis of Egyptian ACoVs reported in this study revealed that five isolates (MF034379.1 ACoV/quail/Sharqia-Egypt/VRLCU-8/2016, MF034380.1 ACoV/cattle egret/Qalubia-Egypt/VRLCU-9/2016, MF034381.1 ACoV/cattle egret/Kafr El Sheikh-Egypt/VRLCU-10/2016, MF034382.1 ACoV/teal/Benisuef-Egypt/VRLCU-11/2016 and MF034383.1 ACoV/cattle egret/Kafr El Sheikh-Egypt/VRLCU-12/2016) were closely related to Egy/Variant 1 of IS/1494/06 genotype origin and very close to the ancestral Egyptian virus (JX174183.1 Egypt/Beni-Suef/01) and previously isolated Egyptian viruses (ck/Eg/BSU-1,4,5/2011) of IS/1494/06 genotype origin (Figure 4). Whilst the majority of isolates (*n* = 9) (MF034372.1 ACoV/house sparrow/Sharqia-Egypt/VRLCU-1/2016, MF034373.1 ACoV/teal/Sharqia -Egypt/VRLCU-2/2016, MF034374.1 ACoV/teal/Dakahlia -Egypt/VRLCU-3/2016, MF034375.1 ACoV/teal/Gharbia-Egypt/VRLCU-4/2016, MF034376.1 ACoV/teal/Qalubia-Egypt/VRLCU-5/2016, MF034377.1 ACoV/quail/Gharbia-Egypt/VRLCU-6, MF034378.1 ACoV/cattle egret/Kafr El Sheikh-Egypt/VRLCU-7/2016, MF034384.1 ACoV/teal/Kafr El Sheikh-Egypt/VRLCU-13/2016 and MF034385.1 ACoV/cattle egret/Menofia-Egypt/VRLCU-14/2016) were clustered within the Egy/Variant 2 subgroup and other Egyptian viruses (ck/Eg/BSU-2,3,6/2011) of IS/885-00 genotype origin (Figure 3, vertical tree, right panel and Figure 4). Taken together, Egy/variant 1 and Egy/variant 2 groups were phylogenetically subclassified into subgroups/subclusters (Figure 4) based on ACoVs emergence, origin, and evolution as a result of extensive diversity and evolution which are characteristic features for most of RNA viruses especially ACoVs.

### 3.4. Selective Pressure Sites and Recombination Analysis

A pairwise comparison bioinformatics approach (SNAP) was applied to determine the synonymous and non-synonymous substitution rates and selective evolutionary pressure for the S1 protein. The results presented in Figure 5 indicate the selection trends where numbers above zero indicate the positive selection, around zero shows the neutral selection, and below zero indicates the negative or purifying selection. The selection profiles of the amino acid sequence of all fourteen Egyptian ACoV strains showed different patterns within the S1 protein. Analysis of subtract synonymous and non-synonymous substitution rates in the isolates also demonstrated the presence of purifying (negative) selection. The cumulative difference between the non-synonymous substitution rate (dN) and the synonymous substitution rate (dS) (i.e., dN-dS) revealed that positive selection along the spike (S1) protein while negative and neutral selection was also detected (Figure 5). When the S1 genes of ACoV strains were used as the query, a number of possible recombination sites were identified. Recombination analysis for the aligned sequences under study report the detection of possible recombination events; (1) AY091552.2 IBV IS/720/99/S and JX173489.1 IBV/Eg/CLEVB-1/012 strains to produce a recombinant strain MF034384.1 ACoV/teal/Kafr El Sheikh-Egypt/VRLCU-13/2016 (Figure 6A); (2) KC197206.1 IBV/ck/Egypt/12vir6109-79/2012 and JX174186.1 IBV Ck/Eg/BSU-3/2011 strains to produce a recombinant strain MF034381.1 ACoV/cattle egret/Kafr El Sheikh-Egypt/VRLCU-10/2016 (Figure 6B).

## 4. Discussion

Continuous generations of natural mutant viruses or quasispecies subpopulations may result in the adaptation of such viruses to evade the host immune system and to gain higher transmissibility to new hosts [28]. ACoVs are continuously evolving, and the variant strains are one of the major emerging problems in the Egyptian poultry industry for the past five years and have resulted in significant economic losses. Furthermore, the infection predisposes the host for secondary bacterial infections resulting in an even higher morbidity and mortality rates [29]. Different serotypes have been reported worldwide and new variant serotypes are continuously being recognized [4]. However, there is no clear classification criterion for the circulating ACoVs, worldwide, which possesses complications in establishing epidemiological linking. Previous molecular studies revealed that the S1 subunit of S gene of ACoV is responsible for determining its serotypes [30] and new ACoV genotypes could arise as a result of substitutions in the amino acids of the S1 subunit [3]. S1 subunit has three HVRs, located within amino acids 38–67, 91–141, and 274–387 [3,4,5] and these HVRs were considered an essential determinant of coronavirus serotype specificity [31].

Factors that contribute in the evolution and emergence of ACoV in commercial poultry are largely unknown. Evidences are that wild birds might constitute transmission vectors for ACoVs even in a huge geographic territory. It has also been proposed that when an ACoV infects the same wild bird host, this could lead to the emergence of new variants of ACoV [32] and the possibility of recombination between different ACoV strains. To assess the ACoV existence and carrying-potential of Egyptian wild birds for the first time in the country, 557 cloacal swabs were randomly collected from different wild bird species in different Egyptian governorates. Detection of gamma-coronaviruses (ACoVs) in wild bird samples from geographically separated populations highlights the potential of wild birds in the possible dissemination of ACoV. Since most of the ACoV positive wild birds were migratory and that Egypt has been proposed to be an important gateway in the Middle East and North of Africa for most of the emerging infectious diseases, it is likely that the mixing of birds and viruses from different hemispheres can lead to disease transmission. From this and other studies, it is reasonable to assume that a great variety of hitherto undetected ACoVs exist in wild bird species. Previous studies that examined the host range and genetic diversity of CoVs have revealed that ACoVs in wild birds are present mainly in wildfowl (Anseriformes) and waders (Charadriiformes) [22,33,34]. Our results corroborate these findings and indicate that ACoVs are common among different wild birds. Conclusively, the data show that there is circulation of genetically divergent ACoVs among wild bird populations in geographically distinct areas of Egypt. Although the number of samples analyzed in this study was limited, it can be assumed that the genetic variation of ACoVs among wild birds is much higher than previously thought. Isolates of ACoV that are sequenced and reported here not only show a high similarity with isolates from Israeli, Libya and Oman strains but are also highly similar (99–100%) to the strains previously reported from commercial poultry flocks, which implies that certain strains may have the potential to spread directly or indirectly to distant regions or to other countries.

Although no discrete and globally accepted classification system exists for ACoV, our results revealed that most of ACoVs available on GenBank are divided into variant and classic groups based on the sequence of the full length S1 genes. The classic group has only the Massachusetts serotype (mass-like/vaccine strains) (GI-1 lineage) according to Valsastro et al. [8]. In 2001, an Egyptian variant I strain (Egypt/Beni-Suef/01), closely related to the Israeli variant strain, was reported on different poultry farms [18]. Another variant was also identified in 2011 [27] and designated as Egyptian variant II. Variant genotype can be subdivided into at least eight subgenotypes according to their origin and evolution; 4/91 genotype, Taiwan and China like, Australia variant, QX variant, IS/1494/06 (Egy/Variant-1) and IS/885 (Egy/Variant-2). Egy/Variant I and II are belonging to GI-23 lineage, which represents a unique geographic wild-type cluster confined to the Middle East according to Valsastro et al. [8]. Some have become dominant in the majority of farms and are involved in respiratory and renal pathologies in the last five years.

However, previous studies reported that the variant group (GI-23 lineage) represents a unique geographic wild-type cluster, indigenous and predominate within the Middle East that includes IS/885/00 and IS/1494/06 [8,18,35], recent studies reported that the majority of the Egyptian ACoV strains (IBVs) belong to QX-like, 4/91, IS/1494/06 (Egy/Variant-1) and IS/885 (Egy/Variant-2) genotypes [27,36]. These genotypes were formerly present in the Egyptian commercial farms five years ago with increasing the mortality rates in broiler flocks but had not been described before in wild birds because of low surveillance and epidemiological studies directed towards wild birds worldwide. Our results indicated the prevalence of Egy/variant 1 (IS/1494/06 genotype) (5/14; 35.7%), in spite of relative abundance of Egy/variant 2 (IS/885 genotype) (9/14; 64.3%). The emergence of Egy/variant 1 (IS/1494/06-like) and Egy/variant 2 (IS/885-like) in Egypt has been introduced to the poultry population raises several speculations; it could be due to the introduction of carriers during importations of chicks from Middle East countries where this genotype was endemic or due to the role of migratory birds as a source of infection [19] that has been proposed in this study. Likewise, clustering of some ACoVs isolated from wild and commercial birds in Libya, Oman, and Kurdistan and allocated within the same Egyptian clusters either Egy/variant 1 and or Egy/variant 2 highlights the role of wild birds for transmission of such viruses. These results indicate the circulation of ACoVs in Egyptian wild birds, which could potentially be through the wild birds of neighboring countries and support the possible role of wild birds in the transmission of such viruses.

Recombination has been well reported and is thought to be a contributing factor in the emergence and evolution of different coronavirus genotypes as well as different species of coronavirus [37]. Thor and colleagues reported that the principal mechanisms for generating genetic and antigenic diversity within ACoV indicate a progressive evolutionary change that is the result of recombination events in the ACoV genome that play a major role in the origin and adaptation of the virus, leading to the emergence of new IB genotypes and serotypes [38]. Many recombination events have been reported in different ACoV strains, not only between field (wild-type) and vaccine viruses but also among field viruses either within the same genotype (intra-genotypic) or between different genotypes (inter-genotypic) [39,40], giving rise to new ACoV genotypes [41]. Recombination analysis for the aligned sequences under study report the detection of recombination events; 1) AY091552.2 IBV IS/720/99/S and JX173489.1 IBV/Eg/CLEVB-1/012 strains to produce a recombinant strain MF034384 ACoV/Teal/Kafr El Sheikh-Egypt/VRLCU-13/2016; 2) KC197206.1 IBV/ck/Egypt/12vir6109-79/2012 and JX174186.1 IBV Ck/Eg/BSU-3/2011 strains to produce a recombinant strain MF034381 ACoV/Cattle erget/ Kafr El Sheikh -Egypt/VRLCU-10/2016, which may be a precursor for further variants. Further studies are required to determine the pathobiological and clinical features of this virus in a chicken model. In addition, continuous sequence analysis for the currently circulating IBVs is highly recommended to study the possible spread of this virus and for better understanding of its phylogenetic relatedness to other viruses. These results indicate a progressive recombination of events that occur among wild birds as well as between the circulating variant field strains. Positively selected fragments of genes encoding viral proteins exposed on the surface of the capsid have been documented in other viruses [42,43]. There is an association between positively selected sites along the S1 subunit identified in this study and mapped neutralizing epitopes. It has been reported that mutations in the S1 protein often result in changes in antigenicity [5]. Likewise, parts of the hypervariable regions (HVR1, 2 and 3) defined in this study were shown to be under strong positive selection in the ACoV strains. Taken together, the strong positively selected motifs among the S1 protein may thus be associated with the immune response and receptor binding and would thus be important in future ACoV vaccine development.

## 5. Conclusions

This is the first report confirming the detection of ACoV in wild birds in different Egyptian governorates. Phylodynamic and evolutionary analysis indicate that wild bird ACoV strains are closely related to the infectious bronchitis viruses (IBVs) reported from commercial poultry, indicating their possible threat to commercial poultry. Intensified surveillance of wild birds is an important means of assessing the relative prevalence of ACoV variants, and this knowledge would aid risk assessments and risk management of these viruses. Wild bird surveillance that includes virus isolation may also be a tool for obtaining strains of ACoVs that can be used for vaccine development and diagnostics. Moreover, this study provides insights into the genetic diversity of ACoVs in wild birds’ reservoirs. Further assessing the prevalence and effects of ACoV infections in wild birds will increase our knowledge on CoV interactions with their hosts and may suggest as yet unexploited avenues for combating CoV infections. There is a clear need for a better understanding of ACoV ecology through full genome sequencing that will provide detailed information on different subgroups, and this would require comprehensive data through better surveillance of wild birds.

## Figures and Tables

**Figure 1 viruses-11-00057-f001:**
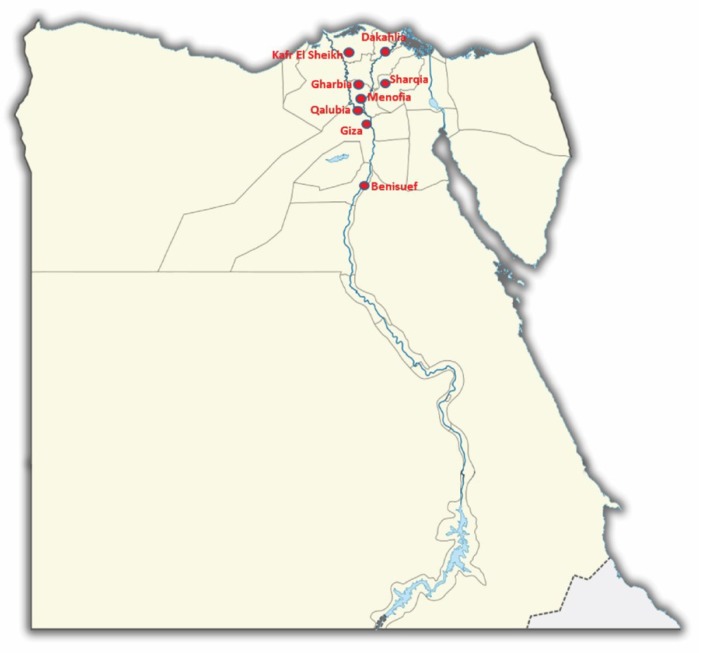
Map of Egypt showing regions (Governorates) of where samples were collected from wild birds.

**Figure 2 viruses-11-00057-f002:**
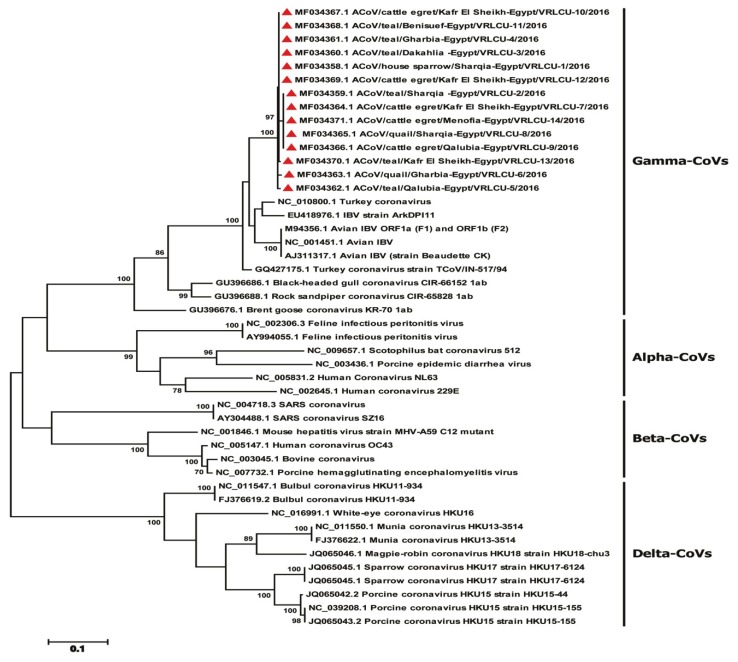
Phylogenetic tree based on the RdRp gene, showing that the detected viruses in this study belonged to gamma-coronaviruses. Trees constructed using maximum likelihood methods. The reported isolates in this study are marked with red triangular.

**Figure 3 viruses-11-00057-f003:**
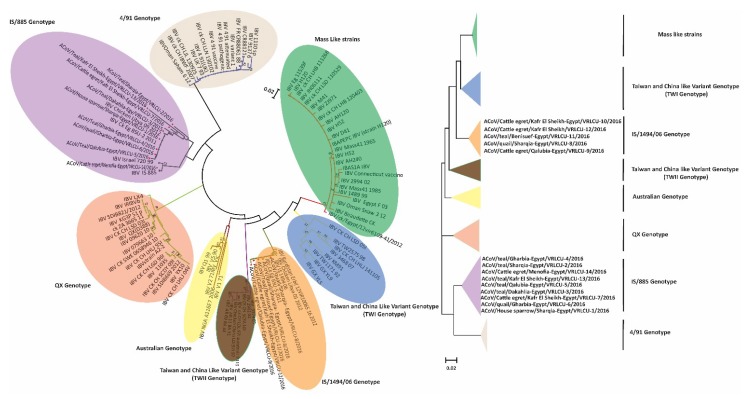
Phylogenetic tree based on S1 gene, showing the relationship between different ACoV genotypes worldwide in relation to ACoV isolates reported in this study (left panel-radiation tree; right panel-vertical tree). The robustness of individual nodes of the tree was assessed using 1000 replications of bootstrap re-sampling of the originally aligned nucleotide sequences. The scale bar represents the number of substitutions per site. The year of isolation and geographical origin of the virus sequences are included in the tree. Trees were constructed using maximum likelihood methods. The reported isolates in this study are marked with red triangle.

**Figure 4 viruses-11-00057-f004:**
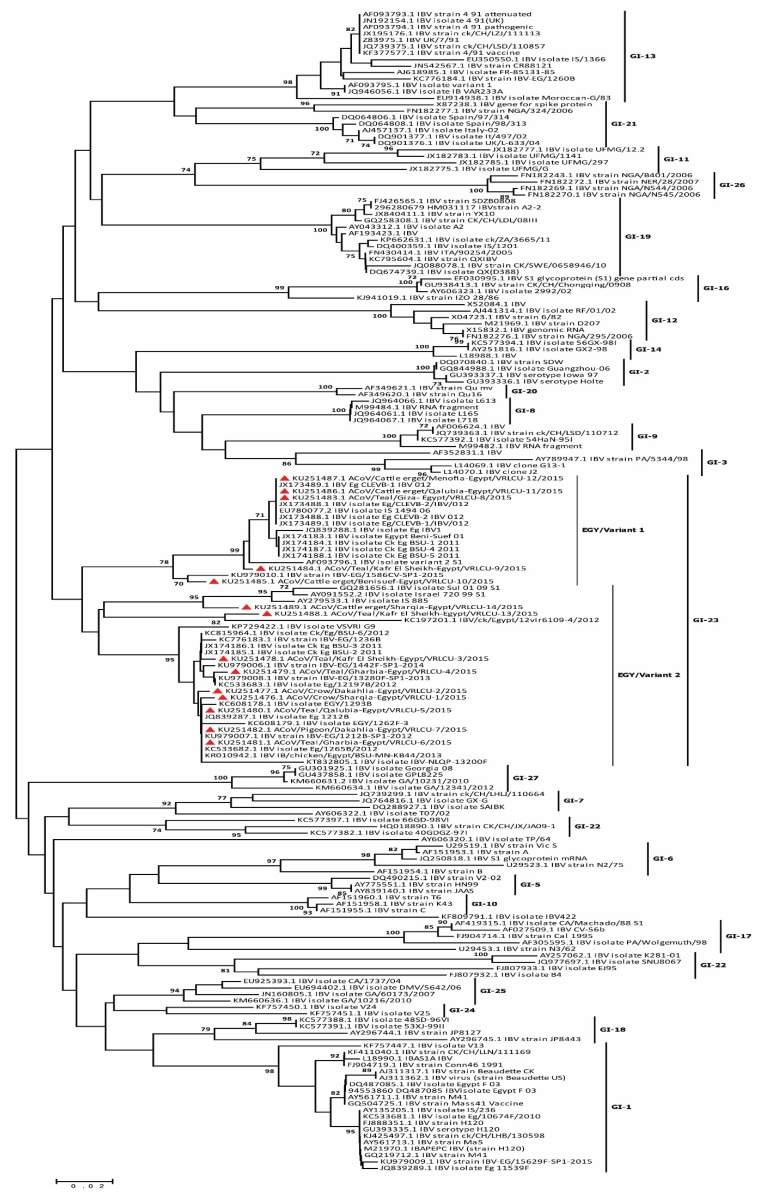
Phylogenetic tree based on a full-length sequence of the S1 gene, showing the relationship between the circulating Egyptian genotypes in commercial poultry sectors in relation to ACoV isolates reported in this study. The robustness of individual nodes of the tree was assessed using 1000 replications of bootstrap re-sampling of the originally aligned nucleotide sequences. Scale bar represents the number of substitutions per site. The year of isolation and geographical origin of the virus sequences are included in the tree. Tree was constructed using maximum likelihood method. The reported isolates in this study are marked with red triangle.

**Figure 5 viruses-11-00057-f005:**
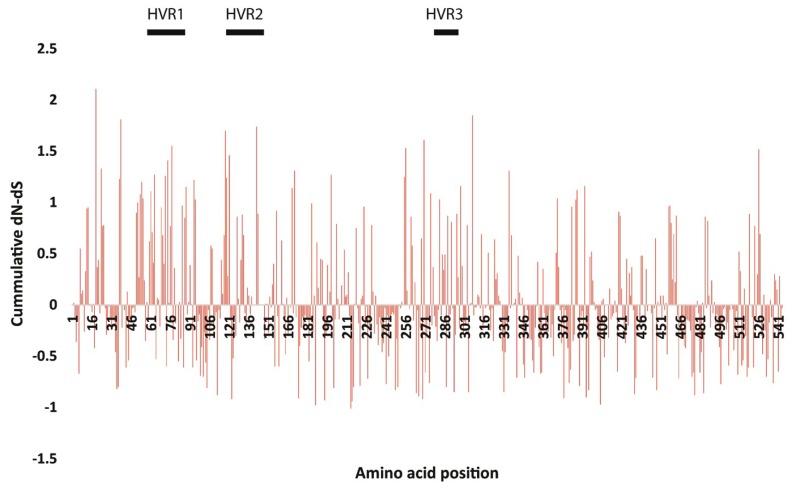
Cumulative dN-dS using SNAP methods along the full length S1 protein of the ACoV strains sequenced in this study.

**Figure 6 viruses-11-00057-f006:**
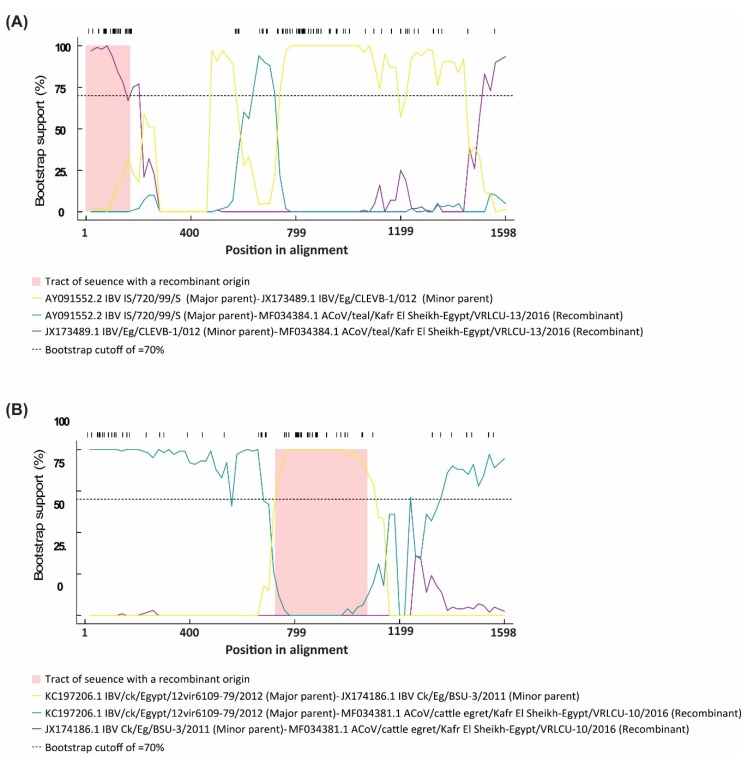
Recombination detection analysis displaying possible recombination events predicted to have occurred in the S1 segment of the (**A**) MF034384.1 ACoV/teal/Kafr El Sheikh-Egypt/VRLCU-13/2016 and (**B**) MF034381.1 ACoV/cattle egret/Kafr El Sheikh-Egypt/VRLCU-10/2016 isolates.

**Table 1 viruses-11-00057-t001:** Overview of wild bird samples involved in the study, and the prevalence of Avian coronaviruses (ACoVs) in different species.

Order	Family	Genus	Species	Positive (*n*)	Sampled (*n*)	Rate%
Passeriformes	Passeridae	*Passer*	*P. domesticus*	1	140	0.7
Galliformes	Phasianidae	*Coturnix*	*C. coturnix*	2	136	1.5
Anseriformes	Anatidae	*Anas*	*A. crecca*	7	150	4.7
Pelecaniformes	Ardeidae	*Bubulcus*	*B. ibis*	4	131	3.1
Total = 4	4	4	4	14	557	2.5

**Table 2 viruses-11-00057-t002:** Proportion of sampled wild birds across different Egyptian provinces.

Species	Sharqia	Dakahlia	Kafr El Sheikh	Gharbia	Qalubia	Menofia	Giza	Benisuef	Total
*C. coturnix*	23	21	20	22	23	15	7	9	140
*P. domesticus*	22	20	23	18	17	14	12	10	136
*A. crecca*	23	22	19	21	20	18	13	14	150
*B. ibis*	23	20	18	21	19	11	9	10	131
Total	91	83	80	82	79	58	41	43	557
Positive (n)	3	1	4	2	2	1	0	1	14
Rate %	3.3	1.2	5	2.4	2.5	1.7	0	2.3	2.5

(n): means number.

**Table 3 viruses-11-00057-t003:** Percentage nucleotide identity among reported viruses in this study.

Sequence	1	2	3	4	5	6	7	8	9	10	11	12	13	14
1	ID	100%	99%	99%	99%	99%	99%	92%	92%	92%	92%	92%	98%	98%
2	100%	ID	99%	99%	100%	99%	99%	92%	92%	92%	92%	92%	98%	98%
3	99%	99%	ID	99%	99%	99%	99%	92%	92%	92%	92%	92%	98%	98%
4	99%	99%	99%	ID	99%	99%	99%	92%	92%	92%	92%	92%	98%	98%
5	99%	100%	99%	99%	ID	99%	99%	92%	92%	93%	92%	92%	98%	98%
6	99%	99%	99%	99%	99%	ID	100%	92%	92%	93%	92%	92%	98%	98%
7	99%	99%	99%	99%	99%	100%	ID	92%	92%	93%	92%	92%	98%	98%
8	92%	92%	92%	92%	92%	92%	92%	ID	100%	99%	100%	100%	92%	92%
9	92%	92%	92%	92%	92%	92%	92%	100%	ID	99%	100%	100%	92%	92%
10	92%	92%	92%	92%	93%	93%	93%	99%	99%	ID	99%	99%	93%	92%
11	92%	92%	92%	92%	92%	92%	92%	100%	100%	99%	ID	100%	92%	92%
12	92%	92%	92%	92%	92%	92%	92%	100%	100%	99%	100%	ID	92%	92%
13	98%	98%	98%	98%	98%	98%	98%	92%	92%	93%	92%	92%	ID	98%
14	98%	98%	98%	98%	98%	98%	98%	92%	92%	92%	92%	92%	98%	ID

^1^MF034372.1 ACoV/house sparrow/Sharqia-Egypt/VRLCU-1/2016, ^2^MF034373.1 ACoV/teal/Sharqia -Egypt/VRLCU-2/2016, ^3^MF034374.1 ACoV/teal/Dakahlia -Egypt/VRLCU-3/2016, ^4^MF034375.1 ACoV/teal/Gharbia-Egypt/VRLCU-4/2016, ^5^MF034376.1 ACoV/teal/Qalubia-Egypt/VRLCU-5/2016, ^6^MF034377.1 ACoV/quail/Gharbia-Egypt/VRLCU-6, ^7^MF034378.1 ACoV/cattle egret/Kafr El Sheikh-Egypt/VRLCU-7/2016, ^8^MF034379.1 ACoV/quail/Sharqia-Egypt/VRLCU-8/2016, ^9^MF034380.1 ACoV/cattle egret/Qalubia-Egypt/VRLCU-9/2016, ^10^MF034381.1 ACoV/cattle egret/Kafr El Sheikh-Egypt/VRLCU-10/2016, ^11^MF034382.1 ACoV/teal/Benisuef-Egypt/VRLCU-11/2016, ^12^MF034383.1 ACoV/cattle egret/Kafr El Sheikh-Egypt/VRLCU-12/2016, ^13^MF034384.1 ACoV/teal/Kafr El Sheikh-Egypt/VRLCU-13/2016 and ^14^MF034385.1 ACoV/cattle egret/Menofia-Egypt/VRLCU-14/2016.

**Table 4 viruses-11-00057-t004:** Sequence comparison of hypervariable regions (HVR) amino acid sequences of ACoV isolates with those of the H120 and Ma5 reference strains.

Strain	HVR1 (60-88)	HVR2 (115-140)	HVR3 (275-292)
IBV strain H120	GSSSGCTVGIIHGGRVVNASSIAMTAPSS	YKH--GGCPITGMLQQHSIRVSAMKNGQ	HNETGANPNPSGVQNIQTY
IBV strain Ma5			
house sparrow/Sharqia-Egypt/VRLCU-1	..Q.Q..A.A.YWSKNFS.A.V.....QN	..SSS.S..L...IP..Y..I...R.NS	Y..SN.H..NG..HT.SI.
teal/Sharqia -Egypt/VRLCU-2	..Q.Q..A.A.YWSKNFS.A.V.....QN	..SSS.S..L...IP..Y..I...R.NS	Y..SN.H..NG..HT.SI.
teal/Dakahlia -Egypt/VRLCU-3	..Q.Q..A.A.YWSKNFS.A.V.....QN	..SSS.S..L...IP..Y..I...R.NS	...SN.H..NG..HT.SL.
teal/Gharbia-Egypt/VRLCU-4	..Q.Q..A.A.YWSKNFS.A.V.....QN	..SSS.S..L...IP.YY..I...R.NS	...SN.H..NG..HT.SL.
teal/Qalubia-Egypt/VRLCU-5	..Q.Q..A.A.YWSKNFS.A.V.....QN	..SSS.S..L...IP.YY..I...R.NS	Y..SN.H..NG..HT.SL.
quail/Gharbia-Egypt/VRLCU-6	..Q.Q..A.S.YWSKNFS.A.V.....QN	..SSS.S..L...IP..Y..I...R.NS	Y..SN.H..NG..HT.SL.
cattle egret/Kafr El Sheikh-Egypt/VRLCU-7	..Q.Q..A.S.YWSKNFS.A.V.....QN	..SSS.S..L...IP.YY..I...R.NS	Y..SN.H..NG..HT.SL.
quail/Sharqia-Egypt/VRLCU-8	..G.Q..A.S.YWSKNFT...V.....DT	..SSS.S..L...IP..Y..I...R.NS	T.VSN.S..TG..NT.NI.
cattle egret/Qalubia-Egypt/VRLCU-9	..G.Q..A.S.YWSKNFT...V.....DT	..SSS.S..L...IP..Y..I...R.NS	T.VSN.S..TG..NT.NI.
cattle egret/Kafr El Sheikh-Egypt/VRLCU-10	..G.Q..A.S.YWSKNFT...V.....DT	..NGQ.S..L..LIP.NH..I.....SR	T.VSN.S..TG..NT.NI.
teal/Benisuef-Egypt/VRLCU-11	..G.Q..A.S.YWSKNFT...V.....DT	..NGQ.S..L..LIP.NH..I.....SR	T.VSN.S..TG..NT.NI.
cattle egret/Kafr El Sheikh-Egypt/VRLCU-12	..G.Q..A.S.YWSKNFT...V.....DT	..NGQ.S..L..LIP.NH..I.....SR	T.VSN.S..TG..NT.NI.
teal/Kafr El Sheikh-Egypt/VRLCU-13	..Q.Q..A.A.YWSKNFS.A.V.....QN	..NGQ.S..L..LIP.NH..I.....SR	Y..SN.S..SG..NT.NLF
cattle egret/Menofia-Egypt/VRLCU-14	..E.Q..A.A.YWSKNFS.A.V.....QN	..NGQ.S..L..LIP.NH..I.....SR	Y..SN.S..SG..NT.NLF

A dot indicates an identical amino acid. A dash indicates an amino acid deletion.

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
