# Peer review of "Genetic Diversity and Phylodynamics of Avian Coronaviruses in Egyptian Wild Birds"

_viruses, 2019, doi:10.3390/v11010057_

Round 1

Reviewer 1 Report

This paper reports isolation of coronaviruses from fecal/cloacal swabs from four species of wild birds in eight different governates in Egypt, and bioinformatic analyses. From a total of 557 wild birds, 14 different coronaviruses were isolated and determined to be avian coronavirus (infectious bronchitis virus). The goal of the work was to determine the potential role of wild birds in transmission and evolution of avian coronaviruses. The complete S1 gene sequences of the 14 isolates were determined and compared to those of other IBV isolates to determine relationships and evolutionary patterns. All 14 fell into only two groups, both related to IBV identified in chickens in Egypt and in nearby countries. Evidence of both positive and negative selection pressure on segments of the S1 gene was identified.  Analyses to determine the evolutionary rate and demonstrate recombination events were carried out. The relevance of the results of the recombination analyses to the present study is unclear since none of the putative recombination events identified involved any of the 14 isolates identified in this study. In the discussion, the significance of the results is overstated: “These results indicate progressive evolution of the circulating Egyptian ACoVs in wild birds, which facilitate possible mutations and or recombination events between different serotypes and confirm the role of wild birds in the transmission of such viruses.” The discussion is poorly written, with frequent language errors

1.    There is a discrepancy among sections regarding what samples were used for virus isolation. The Abstract (line 25) and Results section (line 158) mention tracheal samples in addition to cloacal/fecal swabs, but Materials and Methods only state that cloacal/fecal swabs were collected.

2.    The abstract says that important substitutions were identified in the functional domains of the S1 protein, but the manuscript does not state how it was determined whether a substitution was important or not, and functional domains are not defined or mentioned elsewhere in the manuscript.

3.    The identification of glycosylation sites is not a novel result and should not be mentioned in the abstract.

4.    The relevance of the statement in the introduction that coronaviruses pose zoonotic threats to public health is questionable, since the group of coronaviruses considered in this study, gammacoronaviruses, have never been shown to infect mammals.

5.    Line 76: The date of the first description of AcoV in Egypt would be relevant.

6.    It is unclear why reference 19 is cited (in addition to references 6 and 20) for RT-PCR primers used, because reference 19 cites reference 6 for their S1 primers. However reference 19 also lists primers used for sequencing the S1 gene. Were these primers used for sequencing in the present study? Primers used for sequencing are not mentioned in Materials & Methods.

7.    Lines 118-120: The difference between the two datasets, which were then evidently combined (“Both these datasets were aligned”), is unclear. The first dataset includes “all available S1 gene sequences” and the second includes “most of the publicly available S1 sequences from Egypt.” It seems that all the sequences in the second data set would already be present in the first one. Since the description of the second dataset is preceded by “For global and high-level clustering patterns,” it seems that the two data sets served different purposes. Exactly what was done with each data set is not clear. 

8.    Lines 165-168: The prevalence of 0 in Giza governate is ignored as the lowest prevalence rate, and the prevalence of 1.2% in Dakahlia is reported as the lowest.

9.    For the results described in lines 183-192, the identity matrix should be shown, because of the wide range (93-100%) of nucleotide identity. Readers need to be able to see the Individual nucleotide % identities.

 10.  In line 184, the meaning of “the variant genotype” is unclear.  Which specific variant genotype is being referred to?

 11.  GenBank accession numbers without isolate names are used to identify the isolates in lines 186-191, but the isolate names without GenBank accession numbers are used in Fig. 3. 

 12.  Regarding the order of presentation of results, I find it odd that determining that the viruses identified in the wild birds were gammacoronaviruses comes after having shown that the S1 genes were similar to those of other IBVs, which are gammacornaviruses. 

 13.  Fig. 3 (right part) only shows 13 of the 14 isolates. The text on the left part is so small (and is blurry when enlarged) that it cannot be determined whether all 14 are included in the left part.

 14.  Fig. 3: 

a.    The colors for the IS/885 and 4/91 genotypes do not match in the left and right parts of the figure. IS/885 is dark purple in the left part and light purple in the right part. 4/91 is light purple in the left part and gray in the right part. 

b.    What do the black circles or squares (the resolution of the figure is too low to be able to tell whether they are circles or squares) in the left part indicate? This information is lacking in the figure legend. I would think they indicate the isolates from the present study, except that there are two in the Mass group marked in this way.

c.    Is there some other shape that could be used to color the branches of the tree on the left so that the shapes do not overlap? 

d.    The bootstrap values on the tree on the left are impossible to read. Increasing the magnification only results in blurry numbers.

e.    The text discussion of Fig. 3 refers to Egy/variant 1 and Egy/variant 2, terms which do not appear in the figure. The abstract does equate Egy/variant 1 with IS/1494/06 genotype and Egy/variant 2 with IS/885 genotype.

f.     Lines 204-207 indicate that the right panel of Fig. 3 shows a “higher resolution analysis.” This does not appear to be the case. The tree appears to be at a much lower resolution, lacking most of the information that is in the tree on the left. 

g.    The results stated in lines 207-209 are not visible at all in the figure (Fig. 3), which is too small to read anything on, and becomes blurry when the magnification is increased.

 15.  Lines 192-194: “Deduced amino acids analysis were conducted to establish the genetic spectrum, origin, evolution, and the mutation trend analysis for Egyptian ACoVs,” but the only result reported here is that the S1 proteins most of the isolates had the characteristic 17 potential N-linked gycosylation sites. In addition, how many of the isolates had the 17 potential glycosylation sites and which ones did not?

 16.  Regarding the molecular clock analysis (section 3.4):

a.    units are missing for the molecular clock rate. 

b.    Lines 230-233: It is not appropriate to discuss the meaning of “differences” for which the 95% confidence intervals greatly overlap. 

c.    The comparison of “natural conditions” and “vaccine induced virus evolution in domesticated commercial poultry” is not clear. Were the Egyptian strains considered only those from wild birds, or is there no IBV vaccination in Egyptian commercial poultry?  

 17.  Regarding the analysis for positive and negative selection:

a.    Fig. 4 does not show the distribution of negative and positive selection motifs as the figure legend states. It shows the cumulative dN-dS, from which positive and negative selection motifs are inferred. The criteria for designating positive and negative selection motifs are not stated (e.g what values of dN-dS are used as cut-offs). What criterion is used to designate the strong positive selection noted in the text? 

b.    The authors should justify showing the cumulative dN-dS (difference) rather than dN/dS ratios for each position. 

c.    Marking the hypervariable regions in Fig. 4 would make it easier for the reader to ascertain that there is strong positive selection within the HVRs. 

d.    There are clearly regions with strong positive selection outside the HVRs, and there are regions within HVR 2 and HVR3 where the cumulative dN-dS remains less than 0. 

e.    Line 236-237: “showed two general patterns” This does not really give any information. Are there other possibilities besides strong positive selection and negative selection? 

 18.  Regarding the recombination analysis:

a.    Lines 245-246: According to the GenBank entry for KU251485.1, the isolate name ACoV/Cattle erget/Benisuef-Egypt/VRLCU-10/2016 should be ACoV/Cattle erget/Benisuef-Egypt/VRLCU-10/2015 (year is different)

b.    Lines 243-244: The year is also wrong for the isolate with the GenBank accession number KU251488.1. It should be 2015 instead of 2016.

c.    These two GenBank entries for KU251485 and KU251488 are only 314 nt long. Were the recombination analyses carried out on such short sequences?

d.    Lines 243-245: Choice of the words “isolate,” “strains,” and “strain” is unclear. Some isolates are called isolates while others are called strains. Two isolates together are identified as “strains.” For example: “[GenBank accession number] IBV isolate [isolate name] and [GenBank accession number] [isolate name] strains to produce a recombinant strain [GenBank accession number] [isolate name]”

e.    As already noted, the relationship of the results of the recombination analyses to the rest of the paper is unclear. None of the isolates involved are from the set of 14 isolates from wild birds identified in the present work.

 19.  The discussion is especially in need of language improvement. The poor language obscured the meaning. Many sentences are incorrectly written. 

Some examples (of many):

a.    “Emergence of Egy/variant 1 (IS/1494/06- like) and Egy/variant 2 (IS/885-like) in Egypt has been introduced to poultry population raises several speculations”

b.    “Thor and his colleagues reported that, the principal mechanisms for generating genetic and antigenic diversity within ACoV which indicate progressive evolutionary change as a result of recombination events in ACoV genome which plays a major role in the origin and adaptation of the virus leading to emergence of new IB genotypes and serotypes.”

 20.  Line 271-273: Reference 30 proposed the possibility of recombination between different CoV strains, when they infect the same wild bird host acting as a mixing vessel, as a mechanism for generation of new variants of CoV. Thus the citation belongs at the end of the sentence, since reference 30 proposed both the mixing vessel and recombination. Since IBV does not have a segmented genome, recombination is the only mechanism that could generate new variants in a mixing vessel. Reassortment is not a possibility. So the sentence should not have this mechanism tacked on at the end as a possibility. 

 21.  Line 276. The authors indicate that 0-5% is a “high prevalence” of avian coronavirus in wild birds. What is the criterion for determining if a prevalence is low or high?

 22.  Line 284: Why are only references 31 and 32 included in the discussion of wild bird species in which ACoV has been detected? Why are references 20 and 30 not included?

 23.  Lines 295-297: This sentence is confusing. It implies that they only analyzed HVR sequences and not entire S1 sequences to classify ACoV.

 24.  The sentence in line 302 is redundant with the very long and confusing sentence in lines 306-310. The sentence in line 302 is in the paragraph discussing what their results reveal, and refers to GI-23, which is not labeled in Fig. 3 nor mentioned in the results text. The sentence in lines 306-310 refers to contrasting results from previous studies (“However”). 

 25.  Line 299-300 says the 4/91 genotype is the GI-13 lineage, but lines 306-310 say that 4/91 belongs to GI-23. Perhaps the long confusing sentence in lines 306-310 means to say that GI-23 predominates in the Middle East and that IS/885/00, IS/1494/06, 4/91 and QX genotypes are also present in the Middle East, instead of that they are included in the GI-23 lineage. But IS/1494/06 is included in the GI-23 lineage (according to Valastro et al, from which the GI terminology is derived). 

 26.  Lines 313-315: The basis for declaring 5/14 “low prevalence” and 9/14 “high prevalence” is not clear. Fisher’s exact test shows that 5/14 and 9/14 are not statistically significantly different. 

 27.  Lines 319-321 should make clear that some ACoVs isolated from Libya, Oman, and Kurdistan cluster with ACoVs of both Egy/variant 1 and Egy/variant 2 found in wild birds in Egypt (if that is the case). That they are found in wild birds is important to the point the authors are trying to make.

 28.  Lines 321-324; “These results indicate progressive evolution of the circulating Egyptian ACoVs in wild birds, which facilitate possible mutations and or recombination events between different serotypes and confirm the role of wild birds in the transmission of such viruses.” This overstates the significance of the phylogenetic analysis results just stated, that “some ACoVs isolated from Libya, Oman, and Kurdistan within the same Egyptian clusters either Egy/variant 1 and or Egy/variant 2.” As noted above, wild birds are not even mentioned in the results just stated. Furthermore, the results do not confirm the role of wild birds in transmission. At most they confirm the possibility of a role for wild birds in the transmission of such viruses.

 29.  Lines 336-338: “These results indicate progressive recombination events occurred among wild birds as well as between the circulating vaccine and variant field strains.” Nowhere have the authors made clear that any of the strains they identified being potentially involved in recombination events were vaccine strains.

 30.  Lines 340-341: “There is an association between positively selected sites along the S1 subunit identified in this study and mapped neutralizing epitopes.” Some support is needed for this vague statement. Including the location of mapped neutralizing epitopes on Fig. 3 might provide some support.

 31.  Wrong references cited (examples):

a.    Lines 63-65: A report of identification of one new IBV genotype in one country (China) is cited for “many ACoV variant strains have been isolated causing major problems in the poultry industry, around the globe.” 

b.    Lines 65-67: The paper cited (reference 10) did not provide evidence to individually implicate each of the factors listed in indirect transmission. 

c.    Lines 255-257: The work in reference 26 did not demonstrate nor provide any evidence for adaptation of viruses to evade the host immune system or gain increased transmissibility to new hosts.

d.    Lines 259-260: The work in reference 27 did not address the ability of IBV to predispose the host for secondary bacterial infections nor the effect of these secondary bacterial infections on morbidity and mortality rates. The second sentence in the introduction of reference 27 (as background) does say that infectious bronchitis predisposes the respiratory tract to secondary bacterial infection, but this paper on nephropathogenic IBV does not address secondary bacterial infection at all.

 32.  Some language comments/corrections (not all are listed): 

a.    There are frequent extra articles (“the”)

b.    Line 35: “Taken together, we . . .” The results should be taken together. The authors should not be taken together.

c.    Line 58: wrong word. The authors likely mean “contribute” instead of “attribute.”

d.    Line 72: “well established that” and “known to be” are redundant.

e.    Line 104: It should be “RT-PCR” instead of “PCR”

f.     Line 121: It should be “avoid” instead of “avid.”

g.    Line 133: the URL for the SNAP program is incorrect. It is missing the period before html.

h.    Line 328: “Thor and his colleagues” should be “Thor and her colleagues,” or even better “Thor and colleagues”

 33.  Very minor:

a.    Line 168: Table 2 rather than Table 3 should be referenced here. 

b.    Table 3: Column widths need to be adjusted so that each name of a governate fits on one line and does not have one letter on a second line.

c.    Fig. 2: The second digit in one of the bootstrap values is obscured by a thick vertical bar of the tree.

d.    Reference 29: Title contains extra text at the end.

Author Response

This paper reports isolation of coronaviruses from fecal/cloacal swabs from four species of wild birds in eight different governates in Egypt, and bioinformatic analyses. From a total of 557 wild birds, 14 different coronaviruses were isolated and determined to be avian coronavirus (infectious bronchitis virus). The goal of the work was to determine the potential role of wild birds in transmission and evolution of avian coronaviruses. The complete S1 gene sequences of the 14 isolates were determined and compared to those of other IBV isolates to determine relationships and evolutionary patterns. All 14 fell into only two groups, both related to IBV identified in chickens in Egypt and in nearby countries. Evidence of both positive and negative selection pressure on segments of the S1 gene was identified.  Analyses to determine the evolutionary rate and demonstrate recombination events were carried out. The relevance of the results of the recombination analyses to the present study is unclear since none of the putative recombination events identified involved any of the 14 isolates identified in this study. In the discussion, the significance of the results is overstated: “These results indicate progressive evolution of the circulating Egyptian ACoVs in wild birds, which facilitate possible mutations and or recombination events between different serotypes and confirm the role of wild birds in the transmission of such viruses.” The discussion is poorly written, with frequent language errors

Thank you very much for your time and comments, much appreciated. Thanks for raising comments that certainly helped us to improve the manuscript. As you suggested, we have now re-phrased sentences that depict ambiguities, improved language and each of your additional comment is addressed below:

1.    There is a discrepancy among sections regarding what samples were used for virus isolation. The Abstract (line 25) and Results section (line 158) mention tracheal samples in addition to cloacal/fecal swabs, but Materials and Methods only state that cloacal/fecal swabs were collected.

Apologize for this confusion, we have collected and processed only non-invasive cloacal swabs. This has now been clarified throughout the manuscript.

2.    The abstract says that important substitutions were identified in the functional domains of the S1 protein, but the manuscript does not state how it was determined whether a substitution was important or not, and functional domains are not defined or mentioned elsewhere in the manuscript.

The importance of substitutions was extracted from published reports. However, to make it clearer for the readers, we have not edited these sections and we hope the provided information is fluent and not confusing.

3.    The identification of glycosylation sites is not a novel result and should not be mentioned in the abstract.

Thanks for the suggestion and now it has been removed from the abstract.

4.    The relevance of the statement in the introduction that coronaviruses pose zoonotic threats to public health is questionable, since the group of coronaviruses considered in this study, gammacoronaviruses, have never been shown to infect mammals.

We fully understand the concern and agree that since this article is on avian coronaviruses, which are not isolated from human ever, therefore, we have not rephrased this section for clarity.

5.    Line 76: The date of the first description of AcoV in Egypt would be relevant.

The requested information is now added.

6.    It is unclear why reference 19 is cited (in addition to references 6 and 20) for RT-PCR primers used, because reference 19 cites reference 6 for their S1 primers. However reference 19 also lists primers used for sequencing the S1 gene. Were these primers used for sequencing in the present study? Primers used for sequencing are not mentioned in Materials & Methods.

We have now removed redundancy and we hope it is clear now.

7.    Lines 118-120: The difference between the two datasets, which were then evidently combined (“Both these datasets were aligned”), is unclear. The first dataset includes “all available S1 gene sequences” and the second includes “most of the publicly available S1 sequences from Egypt.” It seems that all the sequences in the second data set would already be present in the first one. Since the description of the second dataset is preceded by “For global and high-level clustering patterns,” it seems that the two data sets served different purposes. Exactly what was done with each data set is not clear. 

Thanks for pointing it out; we had generated two datasets earlier to construct Egyptian-specific tree in addition to the rest of world isolates. However, the submitted version had omitted Egyptian-specific tree. We have now rephrased it to remove any ambiguity and we hope it is acceptable now.

8.    Lines 165-168: The prevalence of 0 in Giza governate is ignored as the lowest prevalence rate, and the prevalence of 1.2% in Dakahlia is reported as the lowest.

We are sorry for this confusion, it is edited now.

9.    For the results described in lines 183-192, the identity matrix should be shown, because of the wide range (93-100%) of nucleotide identity. Readers need to be able to see the Individual nucleotide % identities.

Thanks for the comment; we have now provided this analysis in the manuscript as Table.

Seq.

1

2

3

4

5

6

7

8

9

10

11

12

13

14

1

ID

100%

99%

99%

99%

99%

99%

92%

92%

92%

92%

92%

98%

98%

2

100%

ID

99%

99%

100%

99%

99%

92%

92%

92%

92%

92%

98%

98%

3

99%

99%

ID

99%

99%

99%

99%

92%

92%

92%

92%

92%

98%

98%

4

99%

99%

99%

ID

99%

99%

99%

92%

92%

92%

92%

92%

98%

98%

5

99%

100%

99%

99%

ID

99%

99%

92%

92%

93%

92%

92%

98%

98%

6

99%

99%

99%

99%

99%

ID

100%

92%

92%

93%

92%

92%

98%

98%

7

99%

99%

99%

99%

99%

100%

ID

92%

92%

93%

92%

92%

98%

98%

8

92%

92%

92%

92%

92%

92%

92%

ID

100%

99%

100%

100%

92%

92%

9

92%

92%

92%

92%

92%

92%

92%

100%

ID

99%

100%

100%

92%

92%

10

92%

92%

92%

92%

93%

93%

93%

99%

99%

ID

99%

99%

93%

92%

11

92%

92%

92%

92%

92%

92%

92%

100%

100%

99%

ID

100%

92%

92%

12

92%

92%

92%

92%

92%

92%

92%

100%

100%

99%

100%

ID

92%

92%

13

98%

98%

98%

98%

98%

98%

98%

92%

92%

93%

92%

92%

ID

98%

14

98%

98%

98%

98%

98%

98%

98%

92%

92%

92%

92%

92%

98%

ID

1MF034372.1 ACoV/house sparrow/Sharqia-Egypt/VRLCU-1/2016, 2MF034373.1 ACoV/teal/Sharqia -Egypt/VRLCU-2/2016, 3MF034374.1 ACoV/teal/Dakahlia -Egypt/VRLCU-3/2016, 4MF034375.1 ACoV/teal/Gharbia-Egypt/VRLCU-4/2016, 5MF034376.1 ACoV/teal/Qalubia-Egypt/VRLCU-5/2016, 6MF034377.1 ACoV/quail/Gharbia-Egypt/VRLCU-6, 7MF034378.1 ACoV/cattle egret/Kafr El Sheikh-Egypt/VRLCU-7/2016, 8MF034379.1 ACoV/quail/Sharqia-Egypt/VRLCU-8/2016, 9MF034380.1 ACoV/cattle egret/Qalubia-Egypt/VRLCU-9/2016, 10MF034381.1 ACoV/cattle egret/Kafr El Sheikh-Egypt/VRLCU-10/2016, 11MF034382.1 ACoV/teal/Benisuef-Egypt/VRLCU-11/2016, 12MF034383.1 ACoV/cattle egret/Kafr El Sheikh-Egypt/VRLCU-12/2016, 13MF034384.1 ACoV/teal/Kafr El Sheikh-Egypt/VRLCU-13/2016 and 14MF034385.1 ACoV/cattle egret/Menofia-Egypt/VRLCU-14/2016

 10.  In line 184, the meaning of “the variant genotype” is unclear.  Which specific variant genotype is being referred to?

Thanks for the comments, we have now clarified this in the Result section and discussed it in the Discussion section.

 11.  GenBank accession numbers without isolate names are used to identify the isolates in lines 186-191, but the isolate names without GenBank accession numbers are used in Fig. 3.

We have provided the accession numbers, similar to the text in the Figure 3. Corresponding text is edited in the manuscript.

12.  Regarding the order of presentation of results, I find it odd that determining that the viruses identified in the wild birds were gammacoronaviruses comes after having shown that the S1 genes were similar to those of other IBVs, which are gammacornaviruses. 

Thanks for the comment, we have now re-arranged and re-writing these section to avoid confusion. We hope these are clear now.

 13.  Fig. 3 (right part) only shows 13 of the 14 isolates. The text on the left part is so small (and is blurry when enlarged) that it cannot be determined whether all 14 are included in the left part.

The tree has been now reconstructed with better resolution and we have marked these isolated a red square to facilitate reading.

 14.  Fig. 3: 

a.       The colors for the IS/885 and 4/91 genotypes do not match in the left and right parts of the figure. IS/885 is dark purple in the left part and light purple in the right part. 4/91 is light purple in the left part and gray in the right part.

It has been now recoloured to the same.

b.       What do the black circles or squares (the resolution of the figure is too low to be able to tell whether they are circles or squares) in the left part indicate? This information is lacking in the figure legend. I would think they indicate the isolates from the present study, except that there are two in the Mass group marked in this way.

Figure is now edited with better resolution. We have also mentioned in the figure legends for clarity.

c.    Is there some other shape that could be used to color the branches of the tree on the left so that the shapes do not overlap? 

Thanks for the comment, colours are now changed to void overlapping and we hope these are clear.

d.    The bootstrap values on the tree on the left are impossible to read. Increasing the magnification only results in blurry numbers.

We have now improved the resolution and we hope it is clear now.

e.    The text discussion of Fig. 3 refers to Egy/variant 1 and Egy/variant 2, terms which do not appear in the figure. The abstract does equate Egy/variant 1 with IS/1494/06 genotype and Egy/variant 2 with IS/885 genotype.

We have made corresponding changes in the Abstract, tree and Discussion. Further to your earlier comments, the nomenclature is now equated and we believe is not clear for readers.

f.     Lines 204-207 indicate that the right panel of Fig. 3 shows a “higher resolution analysis.” This does not appear to be the case. The tree appears to be at a much lower resolution, lacking most of the information that is in the tree on the left. 

The tree has been reconstructed and we hope it provide useful information.

g.    The results stated in lines 207-209 are not visible at all in the figure (Fig. 3), which is too small to read anything on, and becomes blurry when the magnification is increased.

The figure is now reconstructed and we hope it is clear now.

 15.  Lines 192-194: “Deduced amino acids analysis were conducted to establish the genetic spectrum, origin, evolution, and the mutation trend analysis for Egyptian ACoVs,” but the only result reported here is that the S1 proteins most of the isolates had the characteristic 17 potential N-linked gycosylation sites. In addition, how many of the isolates had the 17 potential glycosylation sites and which ones did not?

Amino acids have now further analysed in comparison to the currently used vaccines and are provided in the Table 4.

 16.  Regarding the molecular clock analysis (section 3.4):

a.    units are missing for the molecular clock rate. 

b.    Lines 230-233: It is not appropriate to discuss the meaning of “differences” for which the 95% confidence intervals greatly overlap. 

c.    The comparison of “natural conditions” and “vaccine induced virus evolution in domesticated commercial poultry” is not clear. Were the Egyptian strains considered only those from wild birds, or is there no IBV vaccination in Egyptian commercial poultry?  

The molecular clock analysis has been now removed based on the recommendations of reviewer 2.

 17.  Regarding the analysis for positive and negative selection:

a.    Fig. 4 does not show the distribution of negative and positive selection motifs as the figure legend states. It shows the cumulative dN-dS, from which positive and negative selection motifs are inferred. The criteria for designating positive and negative selection motifs are not stated (e.g what values of dN-dS are used as cut-offs). What criterion is used to designate the strong positive selection noted in the text? 

A pairwise comparison bioinformatics approach (SNAP) was applied to determine the synonymous and non-synonymous substitution rates and selective evolutionary pressure for the S1 protein. The results are presented in the Figure 5, in which numbers above one indicate the positive selection, around one shows the neutral selection, whereas below one indicates the negative or purifying selection.

b.    The authors should justify showing the cumulative dN-dS (difference) rather than dN/dS ratios for each position. 

As requested, it has been now edited within the manuscript.

c.    Marking the hypervariable regions in Fig. 4 would make it easier for the reader to ascertain that there is strong positive selection within the HVRs. 

As suggested, the edits have been made.

d.    There are clearly regions with strong positive selection outside the HVRs, and there are regions within HVR 2 and HVR3 where the cumulative dN-dS remains less than 0. 

The cumulative dN-dS carried out on full length S1 protein as it is the major antigenic protein which the vaccine directed towards, so the highest rates of selection occurred at these regions while other parts from the protein might be exposed to selections.

e.    Line 236-237: “showed two general patterns” This does not really give any information. Are there other possibilities besides strong positive selection and negative selection? 

The primary objective was to determine the selection, however, as you suggested we have now included neutral selection as well.

 18.  Regarding the recombination analysis:

a.      Lines 245-246: According to the GenBank entry for KU251485.1, the isolate name ACoV/Cattle erget/Benisuef-Egypt/VRLCU-10/2016 should be ACoV/Cattle erget/Benisuef-Egypt/VRLCU-10/2015 (year is different)

It has been now provided.

b.    Lines 243-244: The year is also wrong for the isolate with the GenBank accession number KU251488.1. It should be 2015 instead of 2016.

It has been now edited and sorry for this mistake.

c.    These two GenBank entries for KU251485 and KU251488 are only 314 nt long. Were the recombination analyses carried out on such short sequences?

GenBank accession numbers are: MF034372.1- MF034385.1. The recombination analyses carried out on full-length S1 genes only.

d.    Lines 243-245: Choice of the words “isolate,” “strains,” and “strain” is unclear. Some isolates are called isolates while others are called strains. Two isolates together are identified as “strains.” For example: “[GenBank accession number] IBV isolate [isolate name] and [GenBank accession number] [isolate name] strains to produce a recombinant strain [GenBank accession number] [isolate name]”

Thanks for the comment, the strain names are now unified and we hope are clear for readers.

e.    As already noted, the relationship of the results of the recombination analyses to the rest of the paper is unclear. None of the isolates involved are from the set of 14 isolates from wild birds identified in the present work.

These isolates were included in the analysis and now we have discussed them in the results and discussion sections.

19.  The discussion is especially in need of language improvement. The poor language obscured the meaning. Many sentences are incorrectly written. 

Some examples (of many):

a.    “Emergence of Egy/variant 1 (IS/1494/06- like) and Egy/variant 2 (IS/885-like) in Egypt has been introduced to poultry population raises several speculations”

b.    “Thor and his colleagues reported that, the principal mechanisms for generating genetic and antigenic diversity within ACoV which indicate progressive evolutionary change as a result of recombination events in ACoV genome which plays a major role in the origin and adaptation of the virus leading to emergence of new IB genotypes and serotypes.”

Thanks for raising this comment. Now the language has been proof read by a native English speaker and all changes are mentioned as tracked-changes. We hope the manuscript is devoid of linguistic errors.

 20.  Line 271-273: Reference 30 proposed the possibility of recombination between different CoV strains, when they infect the same wild bird host acting as a mixing vessel, as a mechanism for generation of new variants of CoV. Thus the citation belongs at the end of the sentence, since reference 30 proposed both the mixing vessel and recombination. Since IBV does not have a segmented genome, recombination is the only mechanism that could generate new variants in a mixing vessel. Reassortment is not a possibility. So the sentence should not have this mechanism tacked on at the end as a possibility. 

We agree with Reviewer’s this comment and have tried to rephrased the sections and we hope these are now clear for the readers.

 21.  Line 276. The authors indicate that 0-5% is a “high prevalence” of avian coronavirus in wild birds. What is the criterion for determining if a prevalence is low or high?

We have rephrased these rates because there is no clearly defined prevalence and thus may be subjective to state low or high. Thanks for pointing it out and we hope it has now no ambiguity. 

 22.  Line 284: Why are only references 31 and 32 included in the discussion of wild bird species in which ACoV has been detected? Why are references 20 and 30 not included?

References 20 and 30 are now included.

 23.  Lines 295-297: This sentence is confusing. It implies that they only analyzed HVR sequences and not entire S1 sequences to classify ACoV.

Apologize for this confusion; we have now edited this section to make it clear for readers.

 24.  The sentence in line 302 is redundant with the very long and confusing sentence in lines 306-310. The sentence in line 302 is in the paragraph discussing what their results reveal, and refers to GI-23, which is not labeled in Fig. 3 nor mentioned in the results text. The sentence in lines 306-310 refers to contrasting results from previous studies (“However”). 

Thanks for pointing it out, it has been edited within the text and figure.

 25.  Line 299-300 says the 4/91 genotype is the GI-13 lineage, but lines 306-310 say that 4/91 belongs to GI-23. Perhaps the long confusing sentence in lines 306-310 means to say that GI-23 predominates in the Middle East and that IS/885/00, IS/1494/06, 4/91 and QX genotypes are also present in the Middle East, instead of that they are included in the GI-23 lineage. But IS/1494/06 is included in the GI-23 lineage (according to Valastro et al, from which the GI terminology is derived). 

It has been now rephrased within the text to remove any confusion.

 26.  Lines 313-315: The basis for declaring 5/14 “low prevalence” and 9/14 “high prevalence” is not clear. Fisher’s exact test shows that 5/14 and 9/14 are not statistically significantly different. 

Thanks for comments; we have provided a revised section to clearly articulate the true picture of positive samples in the country.

 27.  Lines 319-321 should make clear that some ACoVs isolated from Libya, Oman, and Kurdistan cluster with ACoVs of both Egy/variant 1 and Egy/variant 2 found in wild birds in Egypt (if that is the case). That they are found in wild birds is important to the point the authors are trying to make.

The sentence is now edited with clear writing.

 28.  Lines 321-324; “These results indicate progressive evolution of the circulating Egyptian ACoVs in wild birds, which facilitate possible mutations and or recombination events between different serotypes and confirm the role of wild birds in the transmission of such viruses.” This overstates the significance of the phylogenetic analysis results just stated, that “some ACoVs isolated from Libya, Oman, and Kurdistan within the same Egyptian clusters either Egy/variant 1 and or Egy/variant 2.” As noted above, wild birds are not even mentioned in the results just stated. Furthermore, the results do not confirm the role of wild birds in transmission. At most they confirm the possibility of a role for wild birds in the transmission of such viruses.

This conclusion section has been modified to “These results indicate circulation of ACoVs in Egyptian wild birds, which could potentially be through the wild birds of neighboring countries and support the possible role of wild birds in the transmission of such viruses”

 29.  Lines 336-338: “These results indicate progressive recombination events occurred among wild birds as well as between the circulating vaccine and variant field strains.” Nowhere have the authors made clear that any of the strains they identified being potentially involved in recombination events were vaccine strains.

As mentioned above, we have rephrased this section and we hope it is clear now.

 30.  Lines 340-341: “There is an association between positively selected sites along the S1 subunit identified in this study and mapped neutralizing epitopes.” Some support is needed for this vague statement. Including the location of mapped neutralizing epitopes on Fig. 3 might provide some support.

Thanks for this suggestion. We have now provided a table covering crucial mutations in important sites in the S protein and we hope this will help readers.

 31.  Wrong references cited (examples):

a.    Lines 63-65: A report of identification of one new IBV genotype in one country (China) is cited for “many ACoV variant strains have been isolated causing major problems in the poultry industry, around the globe.” 

Relevant references have been added.

b.    Lines 65-67: The paper cited (reference 10) did not provide evidence to individually implicate each of the factors listed in indirect transmission. 

We change the reference now.

c.    Lines 255-257: The work in reference 26 did not demonstrate nor provide any evidence for adaptation of viruses to evade the host immune system or gain increased transmissibility to new hosts.

We change the reference now.

d.    Lines 259-260: The work in reference 27 did not address the ability of IBV to predispose the host for secondary bacterial infections nor the effect of these secondary bacterial infections on morbidity and mortality rates. The second sentence in the introduction of reference 27 (as background) does say that infectious bronchitis predisposes the respiratory tract to secondary bacterial infection, but this paper on nephropathogenic IBV does not address secondary bacterial infection at all.

We change the reference now.

 32.  Some language comments/corrections (not all are listed): 

a.    There are frequent extra articles (“the”)

The manuscript has been extensively English edited now.

b.    Line 35: “Taken together, we . . .” The results should be taken together. The authors should not be taken together.

It has been now edited.

c.    Line 58: wrong word. The authors likely mean “contribute” instead of “attribute.”

It has been now edited.

d.    Line 72: “well established that” and “known to be” are redundant.

It has been now edited.

e.    Line 104: It should be “RT-PCR” instead of “PCR”

It has been now edited.

f.     Line 121: It should be “avoid” instead of “avid.”

It has been now edited.

g.    Line 133: the URL for the SNAP program is incorrect. It is missing the period before html

It has been now edited.

h.    Line 328: “Thor and his colleagues” should be “Thor and her colleagues,” or even better “Thor and colleagues”

It has been now edited.

 33.  Very minor:

a.    Line 168: Table 2 rather than Table 3 should be referenced here. 

It has been now edited.

b.    Table 3: Column widths need to be adjusted so that each name of a governate fits on one line and does not have one letter on a second line.

It has been now edited.

c.    Fig. 2: The second digit in one of the bootstrap values is obscured by a thick vertical bar of the tree.

It has been now edited.

d.    Reference 29: Title contains extra text at the end.

It has been now edited.

Reviewer 2 Report

The manuscripts by Rohaim et al. reports about the epidemiology and molecular characterization of avian coronaviruses in wild Egyptian birds. The study is complete as it includes epidemiology, phylogenetic analysis, recombination detection, coalescence data and selection pressure analyses. I think the manuscript is very interesting and the results are relevant. The structure of the manuscript is fine, but I have some issues about some of the analyses performed (see below). Furthermore, some English revisions are required, and I think that the quality and the clarity of the results (especially the figures and the genotyping) can be significantly improved. Below my concerns.

1. This is just a suggestion. I think it would make more sense to present the results of the RdRP first, that shows that all viruses belong to the genus beta, and afterwards the S1 sequence analysis, which shows that there are actually differences across isolates. Basically, moving lines 199-201 at the beginning of section 3.2. I would also combine sections 3.2 and 3.3 together (lines 204-7 are basically a repetition of what said in section 3.2).  

2. Figure 2. To provide a complete picture, I would also include deltacoronaviruses in the RdrP tree. Also, the legend of Figure 2 states that ML trees were built as well. I would rather show the ML tree, respect to the NJ one, since it provides a more accurate description of viral phylogenetic history. Finally, was a modeltest performed before tree construction?

3. Figure 3. The text in Figure 3 is unreadable and clades are so small that it is difficult to identify the different groups mentioned in the manuscript (e.g. lines 207-210). It is also very hard to see all the strains mentioned in the text (e.g. those mentioned at line 203). I suggest highlighting those strains in the tree to give an immediate visual reference and provide the full vertical tree as supplementary material, so the reader is offered the opportunity to inspect each branch. Also, the clade on the far right within genotype 4/91 doesn’t seem to belong to that genotype. Minor edits: the 4/91 clade is marked with 2 different colors in the 2 trees; I guess the dots in the clades IS/1494/06 and IS/885-00 indicate the sequences of this study (this should be stated in the figure legend), what do the dots in the Mass clade denote?

4. Bayes analyses. First of all, I would avoid mentioning tMRCA in the M&M (as you never show coalescence results). Secondly, I am wondering if this analysis is required at all. The two evolutionary rates you found are not really that different and the one calculated with Egyptian sequences only may be biased by the number of available sequences. Also, were all Egyptian sequences monophyletic (I guess not as they were in 2 different clades)? If not, calculating the evolutionary rate for the Egyptian sequences alone does not make much sense… The evolutionary rate is never even discussed further on and I think it does not add anything significant to the study.

5. Selection pressure analysis. Was this performed on the Egyptian strains only? As before, I don’t think this makes sense if the sequences are not monophyletic… maybe this should be done with the full alignment used for Figure 3.

6. Recombination analysis. This (lines 241-6 and 332-336) is a very odd way of presenting recombination results and it is also not that clear. I would include in the text the positions of the breakpoints and if the recombinant strains were detected only once. Instead of reporting the entire name of the isolates, I would rather mention if those events were inter- or intra-genotypic and which genotypes they involved. Finally, maybe a visual representation of the recombinants (e.g. bootscan) would show this more clearly.

7. Variant/genotype nomenclature (lines 295-310). This bit is very confusing, epecially when considering what reported earlier in the manuscript. You have variant 1 and variant 2, and within variant 2 you have again variant 1 and variant 2. Besides, suddenly specific genotypes with specific names (e.g. GI-13 lineage) appear but they have never been mentioned before (there is just a tree with different genotype names). If there is some sort of accepted genotype nomenclature/classification, maybe this should be explained in the introduction. If not, maybe you should find a less confusing way of referring to clades, a way that is consistent across the text (and possibly already clear since the results section).

Minor:

- Line 21. A word like “causing” seems more appropriate than “possessing” in this context.

- Line 44. This classification is outdated, deltacoronoavirus is now officially recognized as genus.

- Line 51. The correct form is “encode four major structural proteins”.

- Line 58. The sentence “may attribute to the escape mutants” is a bit weird. Maybe you meant “contribute to the generation of escape mutants”?

- Line 58. Typo, it should be “shows”.

- Line 63. Typo, “after its first identification”.

- Line 70. Typo, “its possible roles”.

- Line 80. “detecting” in this case sounds more appropriate than ‘screening”.

- Line 94. You talk here about cloacal/fecal swabs, while the abstract states that also tracheal swabs were analyzed.

- Line 105. Shouldn’t it be “Thermo Fisher”?

- Line 107. Typo, remove “from”.

- Section 2.2. How big was the RdRP sequenced fragment?

- Section 2.3. Please, provide here the accession numbers of the obtained sequences.

- Section 2.5. Any particular reasons why the strict clock model was chosen upon the relaxed model? Also, was a model test performed?

- Line 157. Typo, it should be “screening”.

- Line 157-8. Typo, it should be something like “2.5% positive rate”.

- Section 3.1 + tables 2 and 3. These is some redundancy in information reported here and I think that one of the 2 tables should be removed from the main text. I’d suggest removing the list of provinces from lines 164-5. Also, Table 3 is poorly informative as it is. Maybe a good solution would be to combine tables 2 and 3 together, including for example number of positives (or % of positives) between parenthesis in table 3 (so you’ll have the info of table 2 included in the last line of Table 3).

- Line 173. Typo, “where samples were collected from wild birds”.

- Lines 192-6. This sentence needs English revisions.

- Lines 207-210. This sentence needs English revisions.

- Line 228. “1.266×10−4 substitutions per site per year”.

- Lines 237-240. This sentence needs English revisions.

- Lines 271-273. This sentence needs English revisions.

- Line 276. I wouldn’t call 0.7-4.7 “high prevalence”.

- Line 277, Typo, “highlights”.

- Lines 278-281. This sentence needs English revisions.

- Line 287. Typo, “among wild bird populations”.

- Lines 287-293. This sentence need some extra punctuation.

- Lines 314-15. These values are not a prevalence, but a relative abundance.

- Lines 315-319. This sentence needs English revisions.

- Lines 328-331. This sentence needs English revisions.

- Lines 34 and 338. The study never mentions any vaccine strains…

- Line 341. Typo, “results”.

Author Response

The manuscripts by Rohaim et al. reports about the epidemiology and molecular characterization of avian coronaviruses in wild Egyptian birds. The study is complete as it includes epidemiology, phylogenetic analysis, recombination detection, coalescence data and selection pressure analyses. I think the manuscript is very interesting and the results are relevant. The structure of the manuscript is fine, but I have some issues about some of the analyses performed (see below). Furthermore, some English revisions are required, and I think that the quality and the clarity of the results (especially the figures and the genotyping) can be significantly improved.

Thanks for your comments and appreciation. We have now improved the language and we hope it is clear for readers.

Below my concerns.

1. This is just a suggestion. I think it would make more sense to present the results of the RdRP first, that shows that all viruses belong to the genus beta, and afterwards the S1 sequence analysis, which shows that there are actually differences across isolates. Basically, moving lines 199-201 at the beginning of section 3.2. I would also combine sections 3.2 and 3.3 together (lines 204-7 are basically a repetition of what said in section 3.2).  

Thanks for your suggestions. I completely agree with you as second Reviewer proposed the same. We have now presented the data as suggested.

2. Figure 2. To provide a complete picture, I would also include deltacoronaviruses in the RdrP tree. Also, the legend of Figure 2 states that ML trees were built as well. I would rather show the ML tree, respect to the NJ one, since it provides a more accurate description of viral phylogenetic history. Finally, was a model test performed before tree construction?

We have now included deltacoronaviruses and reconstructed the tree with ML algorism and presented the tree. We hope it is clear now.

3. Figure 3. The text in Figure 3 is unreadable and clades are so small that it is difficult to identify the different groups mentioned in the manuscript (e.g. lines 207-210). It is also very hard to see all the strains mentioned in the text (e.g. those mentioned at line 203). I suggest highlighting those strains in the tree to give an immediate visual reference and provide the full vertical tree as supplementary material, so the reader is offered the opportunity to inspect each branch. Also, the clade on the far right within genotype 4/91 doesn’t seem to belong to that genotype. Minor edits: the 4/91 clade is marked with 2 different colors in the 2 trees; I guess the dots in the clades IS/1494/06 and IS/885-00 indicate the sequences of this study (this should be stated in the figure legend), what do the dots in the Mass clade denote?

Figure 3 has been edited and the reconstructed tree as suggested by the Reviewer.

4. Bayes analyses. First of all, I would avoid mentioning tMRCA in the M&M (as you never show coalescence results). Secondly, I am wondering if this analysis is required at all. The two evolutionary rates you found are not really that different and the one calculated with Egyptian sequences only may be biased by the number of available sequences. Also, were all Egyptian sequences monophyletic (I guess not as they were in 2 different clades)? If not, calculating the evolutionary rate for the Egyptian sequences alone does not make much sense… The evolutionary rate is never even discussed further on and I think it does not add anything significant to the study.

Based on the reviewer advice (this and the other), the evolutionary information has been removed from the manuscript.

5. Selection pressure analysis. Was this performed on the Egyptian strains only? As before, I don’t think this makes sense if the sequences are not monophyletic… maybe this should be done with the full alignment used for Figure 3.

It has been performed on full alignment with representative strains from each genotype.

6. Recombination analysis. This (lines 241-6 and 332-336) is a very odd way of presenting recombination results and it is also not that clear. I would include in the text the positions of the breakpoints and if the recombinant strains were detected only once. Instead of reporting the entire name of the isolates, I would rather mention if those events were inter- or intra-genotypic and which genotypes they involved. Finally, maybe a visual representation of the recombinants (e.g. bootscan) would show this more clearly.

Thanks for your comments; we have now tried to incorporate your suggestions. We believe the presented tree is now clear and acceptable.

7. Variant/genotype nomenclature (lines 295-310). This bit is very confusing, epecially when considering what reported earlier in the manuscript. You have variant 1 and variant 2, and within variant 2 you have again variant 1 and variant 2. Besides, suddenly specific genotypes with specific names (e.g. GI-13 lineage) appear but they have never been mentioned before (there is just a tree with different genotype names). If there is some sort of accepted genotype nomenclature/classification, maybe this should be explained in the introduction. If not, maybe you should find a less confusing way of referring to clades, a way that is consistent across the text (and possibly already clear since the results section).

Thanks for this valuable comment. In the light of Reviewer 2 comment as well, we have now included the reference for each genotype and also we defined the variants. Therefore, we believe this information is sufficient to understand the analysis.

Minor:

-        Line 21. A word like “causing” seems more appropriate than “possessing” in this context.

-         

This change is now made.

-        Line 44. This classification is outdated, deltacoronoavirus is now officially recognized as genus.

This change is now made.

-        Line 51. The correct form is “encode four major structural proteins”.

This change is now made.

-        Line 58. The sentence “may attribute to the escape mutants” is a bit weird. Maybe you meant “contribute to the generation of escape mutants”?

This change is now made.

-        Line 58. Typo, it should be “shows”.

This change is now made.

- Line 63. Typo, “after its first identification”.

This change is now made.

- Line 70. Typo, “its possible roles”.

This change is now made.

- Line 80. “detecting” in this case sounds more appropriate than ‘screening”.

This change is now made.

- Line 94. You talk here about cloacal/fecal swabs, while the abstract states that also tracheal swabs were analyzed.

Sorry for this, this change is now made.

- Line 105. Shouldn’t it be “Thermo Fisher”?

This change is now made.

- Line 107. Typo, remove “from”.

This change is now made.

- Section 2.2. How big was the RdRP sequenced fragment?

It is partial sequencing 561 bp.

- Section 2.3. Please, provide here the accession numbers of the obtained sequences.

It has been now added and edited.

- Section 2.5. Any particular reasons why the strict clock model was chosen upon the relaxed model? Also, was a model test performed?

It has been removed from the manuscript based on the reviewer’s comments.

- Line 157. Typo, it should be “screening”.

It has been now added and edited.

- Line 157-8. Typo, it should be something like “2.5% positive rate”.

It has been now added and edited.

- Section 3.1 + tables 2 and 3. These is some redundancy in information reported here and I think that one of the 2 tables should be removed from the main text. I’d suggest removing the list of provinces from lines 164-5. Also, Table 3 is poorly informative as it is. Maybe a good solution would be to combine tables 2 and 3 together, including for example number of positives (or % of positives) between parenthesis in table 3 (so you’ll have the info of table 2 included in the last line of Table 3).

Both tables now are combined in one table.

- Line 173. Typo, “where samples were collected from wild birds”.

It has been now modified and edited.

- Lines 192-6. This sentence needs English revisions.

It has been now modified and edited.

- Lines 207-210. This sentence needs English revisions.

We have rephrased the sentences

- Line 228. “1.266×10−4 substitutions per site per year”.

It has been now modified and edited.

- Lines 237-240. This sentence needs English revisions.

It has been now modified and edited.

- Lines 271-273. This sentence needs English revisions.

We have rephrased the sentence.

- Line 276. I wouldn’t call 0.7-4.7 “high prevalence”.

It has been now modified and edited.

- Line 277, Typo, “highlights”.

It has been now modified and edited.

- Lines 278-281. This sentence needs English revisions.

It has been now modified and edited.

- Line 287. Typo, “among wild bird populations”.

It has been now modified and edited.

- Lines 287-293. This sentence needs some extra punctuation.

It has been now modified and edited.

- Lines 314-15. These values are not a prevalence, but a relative abundance.

It has been now modified and edited.

- Lines 315-319. This sentence needs English revisions.

We have rephrased the sentence.

- Lines 328-331. This sentence needs English revisions.

We have rephrased the sentence.

- Lines 34 and 338. The study never mentions any vaccine strains…

We have rephrased the sentence.

- Line 341. Typo, “results”.

It has been now edited.